# The Digital Twin: A Potential Solution for the Personalized Diagnosis and Treatment of Musculoskeletal System Diseases

**DOI:** 10.3390/bioengineering10060627

**Published:** 2023-05-23

**Authors:** Tianze Sun, Jinzuo Wang, Moran Suo, Xin Liu, Huagui Huang, Jing Zhang, Wentao Zhang, Zhonghai Li

**Affiliations:** 1Department of Orthopedics, First Affiliated Hospital of Dalian Medical University, Dalian 116600, Chinassuomoran@163.com (M.S.);; 2Key Laboratory of Molecular Mechanism for Repair and Remodeling of Orthopedic Diseases, Dalian 116000, China

**Keywords:** digital twin, AI, musculoskeletal system, biomechanics, personalized medicine

## Abstract

Due to the high prevalence and rates of disability associated with musculoskeletal system diseases, more thorough research into diagnosis, pathogenesis, and treatments is required. One of the key contributors to the emergence of diseases of the musculoskeletal system is thought to be changes in the biomechanics of the human musculoskeletal system. However, there are some defects concerning personal analysis or dynamic responses in current biomechanical research methodologies. Digital twin (DT) was initially an engineering concept that reflected the mirror image of a physical entity. With the application of medical image analysis and artificial intelligence (AI), it entered our lives and showed its potential to be further applied in the medical field. Consequently, we believe that DT can take a step towards personalized healthcare by guiding the design of industrial personalized healthcare systems. In this perspective article, we discuss the limitations of traditional biomechanical methods and the initial exploration of DT in musculoskeletal system diseases. We provide a new opinion that DT could be an effective solution for musculoskeletal system diseases in the future, which will help us analyze the real-time biomechanical properties of the musculoskeletal system and achieve personalized medicine.

## 1. Introduction

With the aging of populations and increasing attention to non-communicable diseases, the global incidence of musculoskeletal system disease has been increasing annually, which has been both a strong indication of the need for medical treatment and the main cause of days lost from work [1]. It has become a globally common symptom and occurs in all ages. Musculoskeletal system diseases have high incidence and disability rates; thus, it is necessary to conduct more in-depth research on diagnosis, pathogenesis, and treatments [2]. The occurrence of these disorders is often accompanied by biomechanical changes, and biomechanical factors are indispensable in the development and treatment of these disorders [3]. Analyzing biomechanical changes can help us improve the understanding of disease pathogenesis and prevention and help physicians choose appropriate treatment strategies. However, conventional biomechanical research methods have certain defects concerning personality analysis or dynamic responses. Digital twin (DT) technology may provide new ideas for further improving biomechanical analysis methods in the medical field. 

Early DTs were introduced and employed in industrial and manufacturing fields, such as aerospace, DT workshops, etc. [4,5,6]. Currently, the goal of personalized medicine is to offer each patient a unique course of treatment. Artificial intelligence (AI) has been gradually applied in clinics, providing clinicians with a repeatable second opinion and laying the technical groundwork for the realization of DT. On the basis of multi-modal data integration and fusion, it has gained several successful applications in the medical field [7]. Referring to cases of DTs applied in industry and combining characteristics of medicine itself, we used this brand-new technical method to create a DT of the lumbar spine in the first instance [8]. The research was the first case of DT technology applied in regard to the spine, using interactions between the physical and digital worlds to predict the biomechanical parameters of intervertebral discs. In some areas, such as real-time synchronization, faithful mapping, high fidelity, various data integration, and full periodicity, DT has incomparable advantages over conventional methods [9]. It can be further applied in the whole musculoskeletal system to counteract the shortcomings of conventional methods and explain complex biomechanical problems. In the future, high-resolution models will be established and DT will finally become a solution for musculoskeletal system diseases to achieve real-time monitoring and/or precise diagnosis. Herein, we discuss the limitations of conventional biomechanical methods and suggest a potential solution for personalized diagnosis and the treatment of musculoskeletal system diseases. Current applications of DT in musculoskeletal systems were reviewed and we hope that there will be more related research on the musculoskeletal system, providing valuable research data and solutions for the prevention, treatment, and monitoring of diseases. At the same time, it will further help us analyze the real-time biomechanical properties of the musculoskeletal system and realize personalized medicine.

## 2. Biomechanics of Musculoskeletal System

Biomechanics are ubiquitous in the musculoskeletal system and play an indispensable role in the occurrence, development, and treatment of disorders, including spine-related diseases such as lower back and neck pain, joint-related diseases such as osteoarthritis, traumatic diseases such as fractures, and other musculoskeletal diseases such as ventilation and rheumatoid arthritis [10]. The musculoskeletal system supports and protects other parts of the body and promotes movement. This system expresses mechanical properties through intricate interactions between bones and soft tissues [11,12]. Stresses on bones include compressive, tensile, and fluid shear stress. Bone formation and reconstruction under stress occur due to changes in the function of local osteoblasts and osteoclasts [13,14].

Intervertebral disc degeneration is one of the major causes of lower back pain; its biology and mechanics have been implicated as the predominant inductive cause [15]. In addition to the common loading mechanisms, chronic loading has a large destructive effect on the human spine. Adverse outcomes include uneven stress distribution and reduced mechanical stiffness of the vertebral body and appendages, disrupting the original stable biomechanical structure of the spine [15,16]. The load-sharing relationship between the discs and facet joints is very intricate, and much depends on the posture of our spine [17]. The degeneration of discs causes an increase in the transmission of force across the facet joints [18]. When facet tropism (FT) occurs, the angles of the ipsilateral sagittal plane of the facet joint are different, whereby the intervertebral disc stress on the larger angle side increases, and the stress increases with increases in FT values [19]. Biomechanical factors also play an important role in joint diseases. Taking the knee as an example, using biomechanical methods can sufficiently analyze and verify the role of the anterolateral soft tissue in its rotational stability and the change of the internal rotation angle after the corresponding ligament injury [20,21].

## 3. Limitations of Conventional Biomechanical Methods

The anatomy of the human musculoskeletal system is complex and there are multiple mechanisms of bone diseases. Conventional biomechanical methods have shortcomings in some aspects, which inevitably lead to deviations in the human anatomy [22]. The research methods of human biomechanics can be divided into in vivo and in vitro research. In vivo research includes conventional technology based on morphology and imaging systems and implantable sensor technology [23]. In vivo research can better understand some complex structures that are difficult to accurately simulate by computer models so as to better predict the progression of diseases and optimize treatment plans [24]. However, several technical and safety bottlenecks such as the volume of sensor systems and the safety of battery technology should be overcome [25].

In vitro experiments include cell or animal experiments, as well as cadaver studies. However, it is hard to simulate the microenvironment with cell experiments, and results from animal models and tissue cultures have low reproducibility. Additionally, human cadavers are scarce and often subject to ethical constraints, making finite element analysis (FEA) an effective tool for studying orthopedic-related diseases [26,27]. Compared with conventional methods, FEA is based on computer models and can be quantitatively analyzed. It has the advantages of non-invasiveness and repeatability, and can also be combined with big data and deep learning [28]. Medical image analysis based on machine and deep learning is playing an increasingly important role in disease prevention and control [29].

However, the finite element (FE) method for the biomechanical analysis of the musculoskeletal system also has limitations. First, the accuracy of the model parameters has a significant impact on the accuracy of this numerical method for simulation. Additionally, current FE methods are based on the quasi-static force analysis of different attitudes, and the dynamic response of the object is simulated by separating the actions, which makes the dynamic response of human actions difficult to achieve in real-time [30]. The analysis of individualized biomechanical properties is also difficult. There are great differences in the biomechanical properties of bones in different individuals and genders [31,32]. Furthermore, the integration of models of the musculoskeletal system with clinical applications is not well understood or demonstrated. There are many data sources and methods for constructing FE models; for example, biomechanical CT is one of the state-of-the-art methods for testing skeletal biomechanics [33]. It can help monitor responses to treatments and assess surgical risks. However, its effectiveness has only been demonstrated for opportunistic use, and there are individual differences in the effectiveness and robustness of thresholds [34].

## 4. What Is Digital Twin?

Professor Michael Grieves first put forward the concept of DT to describe the life management of productions [35]. In the early stages, it was initially defined by the National Aeronautics and Space Administration as an integrated, multiscale, multi-physics simulation of an as-built vehicle that mirrors the life of the corresponding flying twin in the field of aerospace to predict the lifecycle of aircrafts [4]. Then, wider uses emerged in the engineering and manufacturing fields, including simulation, validation, accreditation, etc. [5]. In industry, it can build a virtual copy of a product or workshop through big data, which can interact in real-time with the deep information and physical space that cannot normally be observed. Tao et al. [6] put forward the concept of a DT workshop driven by fused twin data to realize iterative operation, optimal management, and planning and control of workshops. According to the latest research, DT mainly consists of five dimensions, namely, physical entities, models in the virtual space, fused data, service, and the connections [36]. Figure 1 shows the basic structures and new technologies of DT. The new technologies include big data, AI, cloud computing, 5G, and virtual reality (VR), which act as important interactions of the DT system. At the same time, the advancement and popularization of novel technologies promote the progress and precision of DT. The physical entity performs instructions and collects data via sensor devices. Virtual models are twin models in digital space. DT data are muti-scaled and include the fusion among them. The service and connections improve the diversity and fidelity of the DT model.

In recent years, the idea of DT has attracted widespread focus from medical researchers, and it is believed that the application of DT technology in the field of health has great promise [7]. DT is data-centric and various complex substructures are related to each other through technologies such as machine learning and AI to finally form a DT system [37]. At present, the DT system has made certain progress in the prevention and treatment of cardiovascular, musculoskeletal, and immune system diseases [38,39]. It has also made some breakthroughs in imaging diagnosis and radiotherapy [40,41]. Musculoskeletal system diseases are greatly affected by biomechanical factors; thus, the application of the DT system may be more promising in this area.

## 5. Current Applications of Digital Twin in the Musculoskeletal System

The DT system offers new opportunities and possibilities for innovative development in the medical field. High fidelity and real-time data collection are the biggest advantages of this technology. As the backbone of DT, high fidelity has a significant impact on the quality of simulation [42]. The healthcare applications of digital twins focus on personal health management and precision medicine, in which reliable progress has been made in patient recovery [43], drug development [44], disease treatment [45], etc. However, challenges remain in areas such as standardized modeling, data security, and data fusion [46]. Real-time data collection can realize the real-time monitoring of patients to analyze possible abnormal conditions and select individualized treatment plans according to a patient’s condition. It has already been employed for several applications, such as disease prediction and improving diagnostic accuracy [47,48]. There are many categories of medical information, including human data from different sources. Data from the biological body include computed tomography (CT), MRI, electrocardiogram or other scanning equipment test data, and biochemical data such as blood routine, urine testing, and biological enzymes. There are also virtual simulation data such as disease prediction, surgical simulation, virtual trials, etc. The fusion of these data to produce diagnostic results and treatment options must be further studied. However, the data collection of the human musculoskeletal system is mainly about geometric and biomechanical data, giving the application of DT in the musculoskeletal system great advantages. Table 1 compares the advantages and disadvantages of traditional biomechanical methods and DT techniques in the musculoskeletal system.

We first applied this brand-new technology in the musculoskeletal system [8]. During the construction of the DT system, we used CT images to develop an FE model of the lumbar spine. The scanning conditions were set as follows: 120 kV, 125 mA, layer thickness 0.625 mm, layer spacing 0 mm, scanning around the spiral from top to bottom. The raw data were stored in DICOM format and we used Mimics, 3dMax, CATIA V5, and Unity3D software for hybrid modeling designs. Body movement was calculated by the inverse kinematics algorithm. The average time delay introduced by the real-time computation was approximately 24.70 ms, which included 8.04 ms of calculating time and 16.67 ms of refresh and transmission time of the 60 Hz sensor. It satisfied the real-time requirements of the biomechanical lumbar spine. As far as we know, this was the first DT example used in a human musculoskeletal system. Finally, a 3D virtual reality system was developed with the help of Unity3D software to record the real-time biomechanical performance of the lumber spine, providing a new and effective method of real-time planning in the field of spine treatments. Moreover, we used DT construction technology to establish a DT of a human lumbar spine that dynamically displayed the action status and biomechanical performance in real-time. It faithfully mapped the health information of the lumbar spine, demonstrated the digital representation, and could be predicted through simulation. Early warnings regarding dangerous postures for the human lumbar can help avoid damage to the spine and provide a more accurate and individualized method for the prevention and monitoring of spinal diseases.

Other researchers then constructed DTs based on orthopedic surgery models to provide valuable quantitative information for clinical treatment, optimize treatment methods, and reduce the occurrence of surgical risks [49,50]. Ahmadian et al. constructed a DT system that combined computational fluid dynamics and level-set methods to simulate the motion of the bone cement and bone marrow separation interface [49]. After predicting the cement morphology, the data co-registration algorithm was used to transform the fluid dynamics model into a high-fidelity continuous damage mechanics model that enhanced the vertebral body. It could predict the morphology and fracture response of cement injection within the bone microstructure. Hernigou et al. developed a more accurate model with DT technology to improve the method of assessing the downward axis and determining joint orientation [51]. They used CT data from patients without osteoarthritis to determine the direction of the subtalar joint axis. Subtalar compensation was assessed based on knee angle deformity and the direction of the inferior axis, and the inaccuracy of the manual selection of anatomical markers was greatly reduced. Aubert et al. [52] constructed DT models using patients’ postoperative 3D X-ray images. They simulated 12 conditions in the DT system using four stabilizing methods and three bone healing results. Biomechanical characteristics such as stress distribution in the musculoskeletal system after surgery were assessed by applying a load during a motion process to assess the risk of recurrent fractures. Therefore, we are confident that DT will be paid more attention regarding the prevention and treatment of musculoskeletal diseases in the future.

## 6. Disadvantages and Possible Improvements of DT

The use of DT for the biomechanical analysis of the musculoskeletal system is a further improvement against the limitations of traditional approaches, especially in terms of technical safety and a lack of real-time dynamic analyses. However, verifying the results is potentially a significant difficulty for DT. Current comparison methods for modeling and assessment mainly use cadaveric data or data from previous studies, lacking individualized assessments. In other words, the results depend on the assumptions that have been made already. The effective integration of a large number of different component models is also an urgent problem for the application of DT in the medical field. The accuracy of the simulation partly determines the effectiveness of the technique. Therefore, data collection based on multiple dimensions and the use of experimental data for simulation training or fitting verification will be the main trends of digital simulation in the future. More accurate validation methods are also needed to allow DT to be more widely used. Currently, there are difficulties in obtaining real pressure data of the body during movement processes and fusing different categories of medical information. There is a gap between the results of model calculation and actual situations, and how to reduce this gap is another difficulty for DT. For example, there is a possibility that risks not predicted or estimated during modeling may occur during the process of surgery. Ethical issues are one of the main reasons why DT may be disadvantageous, and privacy seems to be the most remarkable challenge [53,54]. The standardized definition of DT facilitates a more systematic analysis of ethical risks in personalized health [55]. Additionally, the cost of DT health systems can lead to injustice in medical conditions, which may exacerbate existing economic disparities.

In our previous study, DT of the human lumbar spine was designed to dynamically display action status and biomechanical performance in real-time. It could not be commanded to make physical entities work optimally. In further research, technologies such as exoskeleton devices or other actuators may be available in the human body to carry out commands through virtual twins so that the human musculoskeletal system functions more optimally. The development of DT requires the integration of multiple disciplines, including medicine, engineering, and health law. A standard set of treatment protocols or tools can facilitate the internationalization of treatment and ethics while enabling patients to subjectively evaluate approaches. In the future, the accessibility of this approach and the demand for related products will gradually increase. Safety, efficiency, and ethical practice will be critical for the development of DT.

## 7. Discussion

To realize personalized medicine, personalization and patient-centric modeling are crucial [56]. DT provides a quantitative method concerning diseases, and it is considered to be a driving force for personalized medicine [57,58]. It offers a potential solution for personalized diagnosis and the treatment of musculoskeletal system diseases which might revolutionize the development of medicine. There are many categories of medical information, and how to integrate the different categories of data is key to the establishment of DT in medicine. In general, the technologies behind DT are classified into two broad categories: mechanical models that integrate multi-scale information and statistical models that must learn from categories of data [44,59]. Mechanistic models incorporate physiological knowledge and the fundamental laws of physics and chemistry, providing a framework to integrate experimental and clinical data [60]. It can be used in surgical simulation, drug discovery, and regulatory decision making. Statistical models include all mathematical methods for inferring relationships from experimental data, using mathematical rules to extract and optimize combinations of personalized biomarkers [61]. Faster-solving surrogates of models may be developed in the form of machine learning models to solve problems in real-time, which can help solve the technical hurdles during the creation of high-fidelity models [62]. This will help accelerate model development and improve the quality of models through the consideration of feedback from interactions between various sub-processes.

Figure 2 shows the connections between the physical and digital twins during the construction and application of DT. The information in the physical space is acquired by sensors and CT scans at key locations and then imported into the numerical calculation system for analysis and calculation. Human reverse dynamics and biomechanics are used to complete the fusion of various hetero-dimensional data such as human bone movement information, mechanical information, and human spatial position data. Moreover, the AI model is constructed by fused data and training data. Finally, through visualization technology, the calculation results are rendered as high-fidelity digital twins of the human musculoskeletal system. It can provide a credible digital dynamic model to show the biomechanical performance of the musculoskeletal system and provide reference data for subsequent human musculoskeletal medical studies. In the future, we hope to build a DT of the whole musculoskeletal system and collect various data required by sensors via wearable VR devices at important parts of the body and key bone positions (Figure 3). The data will then be processed and imported into the numerical computing system for analysis and calculation, which is primarily accountable for the security of network system, platform, medical data, user personal privacy and information [63]. Then, we will use the data transmission system to encode, read, decode, and write the collected data. Technologies such as access control, anonymous generalization, storage networking technology, and dynamic passwords can help the security system effectively prevent malicious attacks from third parties, data theft, and information tampering [64]. Finally, through the DT display system, computer graphics technology will be used to render the calculation results into a high-fidelity DT of the human musculoskeletal system, providing an intuitive and credible digital dynamic model. The real-time biomechanical performance will be provided on the digital model, which will be highly compatible with a real human musculoskeletal system. It will be able to predict the dangerous moments of our bodies in advance through simulation and prediction so as to avoid damage to the musculoskeletal system and provide valuable research data and solutions for the prevention, treatment, and monitoring of musculoskeletal diseases.

At present, application examples in musculoskeletal systems represent the initial exploration of DT, and there are still many shortcomings and limitations, such as the accuracy of simulations and the integration of data from different sources. Additionally, there is no consensus on the extent to which DTs can transform healthcare systems over the next decade due to technical, regulatory, and ethical barriers [65]. New technologies, such as big data, cloud computing, blockchain, and VR, which act as important interactions of the system, provide more motivation concerning the medical application of DT. The DT system is based on the acquisition and simulation of multi-source data; thus, it is very sensitive to data processing efficiency and required latency. Using laboratory datasets for simulation training or fitting verification will help improve the efficiency of digital simulations. At the same time, modulus-based DT optimization and soft sensor-based efficiency monitoring will help to develop DT systems with better performance. However, this requires a trade-off between the cost of deep learning and the performance of the DT system. Only in this way can both reduce the cost of time and materials, making DT an effective tool for analyzing the musculoskeletal system and human health. In the traditional framework, the dependency structure of each module reflects the dependence between biological processes, which is difficult to modify or extend. The open-source modular software platform for model simulation and integration proposed by Masison et al. can realize distributed model construction and integration and support a decentralized, community-based model construction process [66]. This provides new ideas for the application of DT technology in the medical field. Accordingly, we hope that this article will inspire researchers to pursue more scientific applications of DT in the field of medicine, especially regarding the musculoskeletal system.

## 8. Conclusions

The application of DT will be increasingly popular in the future of healthcare services and will become a new platform for the health management of the musculoskeletal system. Furthermore, DT technology in combination with intelligent medical technologies will have a significant impact on personalized medicine. Simulations are performed based on the DT system of the patient to select more accurate, appropriate, and individualized treatments. In conclusion, it could be the solution for personalized diagnosis and the treatment of musculoskeletal system diseases.

## Figures and Tables

**Figure 1 bioengineering-10-00627-f001:**
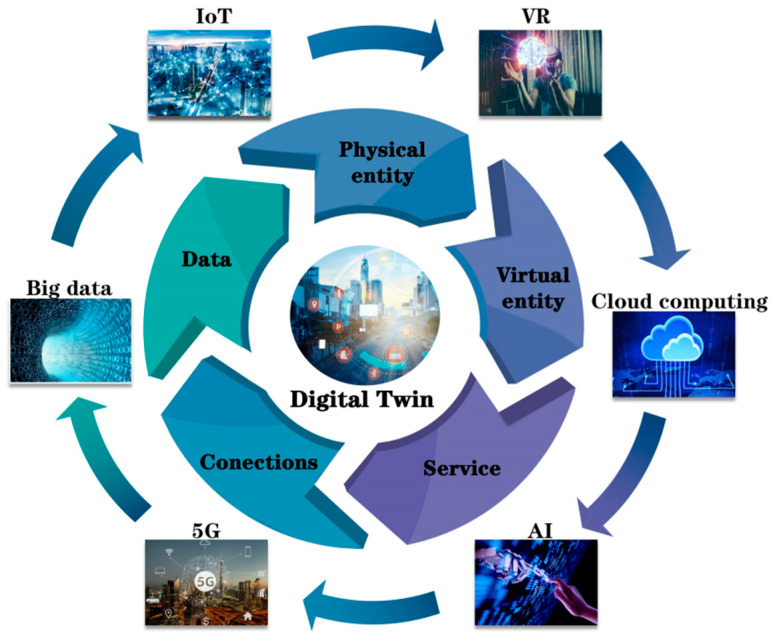
The basic structure and key technologies of digital twin. IoT: Internet of Things; VR: virtual reality; AI: artificial intelligence.

**Figure 2 bioengineering-10-00627-f002:**
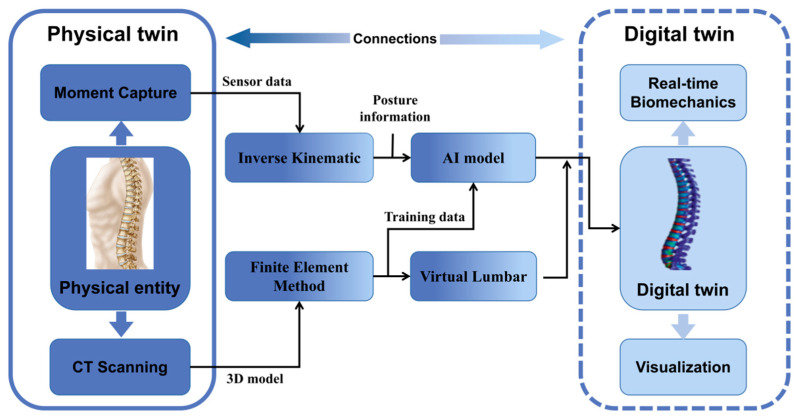
The connections between physical and digital twins. CT = computerized tomography; AI = artificial intelligence.

**Figure 3 bioengineering-10-00627-f003:**
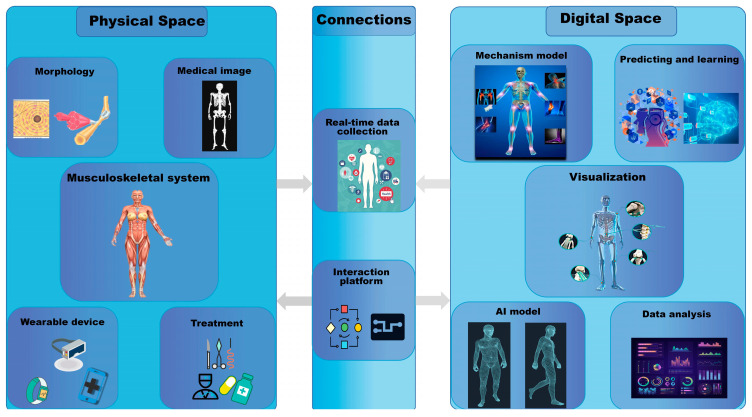
Digital twin of the whole musculoskeletal system for personalized diagnosis and treatment.

**Table 1 bioengineering-10-00627-t001:** Comparison of traditional biomechanical methods and DT techniques in musculoskeletal system.

Method	Advantage	Disadvantage
Morphology	More accurate structural features can be presented based on anatomical and imaging techniques.	Invasive, ethical and safety issues.
Sensers	Quantitative evaluation of biomechanical changes in human body parameters through digital simulations.	Sensor volume and safety issues.
Animal model	Similar to the human body and avoids ethical barriers.	Low reproducibility of animal models and tissue cultures.
Finite element analysis	Reproducible, quantifiable, and non-invasive.	Low simulation accuracy and quasi-static analysis.
Digital twin	Reproducible, quantitative, personalized, and dynamic analysis.	Lack of standardization, high cost, and immature application.

## Data Availability

Not applicable.

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
