# Peer review of "The Digital Twin: A Potential Solution for the Personalized Diagnosis and Treatment of Musculoskeletal System Diseases"

_bioengineering, 2023, doi:10.3390/bioengineering10060627_

Round 1

Reviewer 1 Report (Previous Reviewer 2)

No additional comments

Author Response

Thank you for your recognition of our work.

Reviewer 2 Report (New Reviewer)

I do not have enough knowledge in the field of the use of digital twins in the health field. This article is quite general without referring to all the existing literature, without being in the field, I quickly found other reviews not cited in this document.

Digital Twins in Healthcare: Is It the Beginning of a New Era of Evidence-Based Medicine? A Critical Review, July 2022,Journal of Personalized Medicine 12(8):1255, Patrizio Armeni Irem Polat Leonardo De Rossi Anna Gatti

Developments of digital twin technologies in industrial, smart city and healthcare sectors: a survey, Yang et al. Complex Eng Syst 2021

moreover, I am surprised to obtain a version of the article with modifications in the text not validated....

The subject is interesting, but not seriously enough treated, and the contribution of this work not developed enough. A serious revision is needed.

Author Response

Reviewer 2:

  1. I do not have enough knowledge in the field of the use of digital twins in the health field. This article is quite general without referring to all the existing literature, without being in the field, I quickly found other reviews not cited in this document. Digital Twins in Healthcare: Is It the Beginning of a New Era of Evidence-Based Medicine? A Critical Review, July 2022, Journal of Personalized Medicine 12(8):1255, Patrizio Armeni Irem Polat Leonardo De Rossi Anna Gatti. Developments of digital twin technologies in industrial, smart city and healthcare sectors: a survey, Yang et al. Complex Eng Syst 2021

Response:

We are extremely grateful to reviewer for pointing out the problem. According to the comment, we expanded the search scope and added relevant literature in the revised manuscript.

Details:

    … The healthcare applications of digital twins focus on personal health management and precision medicine, in which reliable progress has been made in patient recovery[43], drug development[44], disease treatment[45], etc. However, challenges remain in areas such as standardized modeling, data security, and data fusion[46].

At present, all application examples in musculoskeletal system are the initial exploration of DT, and there are still many shortcomings and limitations, such as the accuracy of simulations and the integration of data from different sources. Also, there is no consensus on the extent to which DTs can transform healthcare system over the next decade due to technical, regulatory and ethical barriers[65]. We hope that this article will inspire readers to make more scientific applications of DT in the field of medicine.

[43]   Rivera, L.F.; Jiménez, M.; Angara, P.; Villegas, N.M.; Tamura, G.; Müller, H.A.Towards continuous monitoring in personalized healthcare through digital twins. In Proceedings of the Proceedings of the 29th Annual International Conference on Computer Science and Software Engineering, Toronto, Ontario, Canada, 2019; pp. 329–335.

[44].        Corral-Acero, J.; Margara, F.; Marciniak, M.; Rodero, C.; Loncaric, F.; Feng, Y.; Gilbert, A.; Fernandes, J.F.; Bukhari, H.A.; Wajdan, A.; et al. The 'Digital Twin' to enable the vision of precision cardiology. European heart journal 2020, 41, 4556-4564, doi:10.1093/eurheartj/ehaa159.

[45].        Geris, L.; Lambrechts, T.; Carlier, A.; Papantoniou, I.J.C.O.i.B.E. The future is digital: in silico tissue engineering. 2018, 6, 92-98.

[46].        Yang, D.; Karimi, H.R.; Kaynak, O.; Yin, S. Developments of digital twin technologies in industrial, smart city and healthcare sectors: a survey. Complex Engineering Systems 2021, 1, 3, doi:10.20517/ces.2021.06.

[65].  Armeni, P.; Polat, I.; De Rossi, L.M.; Diaferia, L.; Meregalli, S.; Gatti, A. Digital Twins in Healthcare: Is It the Beginning of a New Era of Evidence-Based Medicine? A Critical Review. Journal of personalized medicine 2022, 12, doi:10.3390/jpm12081255.

  1. Moreover, I am surprised to obtain a version of the article with modifications in the text not validated.... The subject is interesting, but not seriously enough treated, and the contribution of this work not developed enough. A serious revision is needed.

Response:

    Thank you very much for your comments. In fact, our manuscript is a resubmission of a revised manuscript based on the comments of previous reviewers. As the reviewer said, DT is a very interesting subject that incorporates a lot of new technologies and has made some progress in the field of healthcare. It is a very clinically significant area of research that should be given full attention. However, the studies of the musculoskeletal system are still initial applications. They provide a good idea for future scientific research and clinical work and we believe that with the gradual improvement of technical means, DT is likely to become a solution for future diseases of the musculoskeletal system. According to the reviewer’s suggestions, we have added corresponding content in many places in the hope of getting the approval of the reviewers. The details are shown in the revised manuscript.

Reviewer 3 Report (New Reviewer)

Thanks for the article it is well written and very new topic, which makes it very interesting. Hopefully more DT related research will be published and further innovations come out.

Author Response

Thank you for your recognition of our work.

Reviewer 4 Report (New Reviewer)

Due to the high prevalence and rates of disability associated with musculoskeletal system diseases, more thorough research into diagnosis, pathogenesis, and treatments is required. One of the key contributors to the emergence of diseases of the musculoskeletal system is thought to be changes in the biomechanics of the human musculoskeletal system. But there are some defects in personal analysis or dynamic responses in current biomechanical research methodologies. Digital twin (DT) is initially known as an engineering concept that reflects the mirror image of a physical entity. With the application of medical image analysis and artificial intelligence (AI), it enters our lives and showed the potential to further applied in medical field.

In this perspective article, the authors provided a new opinion that DT could be an effective solution for musculoskeletal system diseases in the future, which will help them analyze the real-time biomechanical properties of the musculoskeletal system and achieve personalized medicine.

This is an interesting perspective, yet revised and improved, probably with a previous round.

I have some minor suggestions:

1.       The abstract must be improved better summarizing the sections.

2.       A clear purpose is lacking, please add.

3.       Describe figure 1 and 2 in details.

4.       Reference must be recalled with [].

Author Response

Reviewer 4:

Due to the high prevalence and rates of disability associated with musculoskeletal system diseases, more thorough research into diagnosis, pathogenesis, and treatments is required. One of the key contributors to the emergence of diseases of the musculoskeletal system is thought to be changes in the biomechanics of the human musculoskeletal system. But there are some defects in personal analysis or dynamic responses in current biomechanical research methodologies. Digital twin (DT) is initially known as an engineering concept that reflects the mirror image of a physical entity. With the application of medical image analysis and artificial intelligence (AI), it enters our lives and showed the potential to further applied in medical field.

In this perspective article, the authors provided a new opinion that DT could be an effective solution for musculoskeletal system diseases in the future, which will help them analyze the real-time biomechanical properties of the musculoskeletal system and achieve personalized medicine.

This is an interesting perspective, yet revised and improved, probably with a previous round.

I have some minor suggestions:

  1. The abstract must be improved better summarizing the sections.

Response:

We are extremely grateful to reviewer for pointing out the problem. According to the comment, we improved the abstract to better summarize the sections in the revised manuscript.

Details:

    Due to the high prevalence and rates of disability associated with musculoskeletal system diseases, more thorough research into diagnosis, pathogenesis, and treatments is required. One of the key contributors to the emergence of diseases of the musculoskeletal system is thought to be changes in the biomechanics of the human musculoskeletal system. But there are some defects in personal analysis or dynamic responses in current biomechanical research methodologies.  Digital twin (DT) is initially known as an engineering concept that reflects the mirror image of a physical entity. With the application of medical image analysis and artificial intelligence (AI), it enters our lives and showed the potential to further applied in medical field. Consequently, we believe that DT can take a step towards personalised healthcare by guiding the design of industrial personalised healthcare systems. In this perspective article, we discuss the limitations of traditional biomechanical methods and the initial exploration of DT in musculoskeletal system diseases. We provide a new opinion that DT could be an effective solution for musculoskeletal system diseases in the future, which will help us analyze the real-time biomechanical properties of the musculoskeletal system and achieve personalized medicine.

  1. A clear purpose is lacking, please add.

Response:

  Thank you very much for your comments. According to the reviewer, we added the purpose of this article in the section of introduction in the revised manuscript.

Details:

We hope that there will be more DT-related researches on the musculoskeletal system, providing valuable research data and solutions for the prevention, treatment and monitoring of diseases. At the same time, it will further help us analyze the real-time biomechanical properties of the musculoskeletal system and realize personalized medicine.

  1. Describe figure 1 and 2 in details.

Response:

We are extremely grateful to the reviewer for pointing out the problem. According to the comment, we added the description of figure 1 and 2 in the revised manuscript.

Details:

… Figure 1 shows the basic structures and the new technologies of DT. The new technologies include big data, AI, cloud computing, 5G and virtual reality (VR), which act as important interactions of DT system. At the same time, the advancement and popularization of novel technologies promote the progress and precision of DT.

… Figure 2 shows the connections between physical space and the digital space during the construction and application of DT. The information in the physical space is acquired by sensors and CT scans at key locations, and then imported into the numerical calculation system for analysis and calculation. The human reverse dynamics and biomechanics are used to complete the fusion of various heterodimensional data such as human bone movement information, mechanical information, and human spatial position data. Moreover, the AI model is constructed by fused data and training data. Finally, through the visualization technology, the calculation results are rendered as high-fidelity digital twins of human musculoskeletal system. It can provide a credible digital dynamic model to show the biomechanical performance of the musculoskeletal system and provide reference data for subsequent human musculoskeletal medical researches.

  1. Reference must be recalled with [].

Response:

We are extremely grateful to the reviewer for pointing out the problem. According to the comment, we have revised the format of references in the revised manuscript.

Reviewer 5 Report (New Reviewer)

Thank you for the opportunity to review the article entitled "The Digital Twin: A Potential Solution for Personalized Diagnosis and Treatment of Musculoskeletal System Diseases?

This manuscript has been classified in the "Perspective" category. The authors also write in the title "Potential solution". So it may turn out that this tool will not work and will not be widely used in diseases of the musculoskeletal system.

In my opinion, what is missing here is a clearly  defined purpose of this work. I do not fully understand whether this is a review of current scientific reports or an anticipation of the future.

The scientific reports on the use of DT presented so far are poorly described in my opinion.

In the discussion part, the authors rather present the prospects of future activity, and do not discuss the current state of knowledge and advancement of technology, which should be included in this part of the work.

Author Response

Reviewer 5:

  1. This manuscript has been classified in the "Perspective" category. The authors also write in the title "Potential solution". So it may turn out that this tool will not work and will not be widely used in diseases of the musculoskeletal system. In my opinion, what is missing here is a clearly defined purpose of this work. I do not fully understand whether this is a review of current scientific reports or an anticipation of the future.

Response:

We deeply appreciate the suggestions. Digital twins is a very new technology, especially in the field of life and health. At present, the healthcare applications of DT only focus on personal health management and precision medicine, in which reliable progress has been made in cardiovascular disease, surgery, pharmacy, etc. The studies of the musculoskeletal system are still initial applications due to the imperfection of technologies. However, they provided a good idea for future scientific research and clinical work and we believe that with the gradual improvement of technical means, DT is likely to become a solution for future diseases of the musculoskeletal system. In this perspective article, we discuss the limitations of traditional biomechanical methods and the initial exploration of DT in musculoskeletal system diseases. Based on the review of current scientific reports, we provide a new opinion that DT could be an effective solution for musculoskeletal system diseases in the future, which will help us analyze the real-time biomechanical properties of the musculoskeletal system and achieve personalized medicine.  According to the reviewer, we supplemented the literatures and added the content of current state of the technology. The details were shown in the revised manuscript.

  1. The scientific reports on the use of DT presented so far are poorly described in my opinion. In the discussion part, the authors rather present the prospects of future activity, and do not discuss the current state of knowledge and advancement of technology, which should be included in this part of the work.

Response:

Thank you very much for pointing out the problems. According to the reviewer, we supplemented the literatures and added the content of current state of knowledge and advancement of the technology.

Details:

    … The healthcare applications of digital twins focus on personal health management and precision medicine, in which reliable progress has been made in patient recovery[43], drug development[44], disease treatment[45], etc. However, challenges remain in areas such as standardized modeling, data security, and data fusion[46].

At present, all application examples in musculoskeletal system are the initial exploration of DT, and there are still many shortcomings and limitations, such as the accuracy of simulations and the integration of data from different sources. Also, there is no consensus on the extent to which DTs can transform healthcare system over the next decade due to technical, regulatory and ethical barriers[65]. We hope that this article will inspire readers to make more scientific applications of DT in the field of medicine.

The use of DT for biomechanical analysis of the musculoskeletal system is a further improvement against the limitations of traditional approaches, especially in terms of technical safety and lack of real-time dynamic analysis. However, how to verify the results is a potentially huge difficulty for DT. Current comparison methods for modeling and assessment mainly use cadaveric data or data from previous studies, lacking individualized assessments. In other words, the results depend on the assumptions we have made already. The effective integration of a large number of different component models is also an urgent problem for DT in the application of the medical field……

Discussion:

    ……

At present, all application examples in musculoskeletal system are the initial exploration of DT, and there are still many shortcomings and limitations, such as the accuracy of simulations and the integration of data from different sources. Also, there is no consensus on the extent to which DTs can transform healthcare system over the next decade due to technical, regulatory and ethical barriers[65]. In the traditional framework, the dependency structure of each module reflects the dependence between biological processes, which is difficult to modify or extend. The open-source modular software platform for model simulation and integration proposed by Masison et al. can realize distributed model construction and integration, and support a decentralized, community-based model construction process[66]. This provides new ideas for the application of DT technology in the medical field. Accordingly, we hope that this article will inspire researchers to make more scientific applications of DT in the field of medicine, especially the musculoskeletal system.

[43]   Rivera, L.F.; Jiménez, M.; Angara, P.; Villegas, N.M.; Tamura, G.; Müller, H.A.Towards continuous monitoring in personalized healthcare through digital twins. In Proceedings of the Proceedings of the 29th Annual International Conference on Computer Science and Software Engineering, Toronto, Ontario, Canada, 2019; pp. 329–335.

[44].        Corral-Acero, J.; Margara, F.; Marciniak, M.; Rodero, C.; Loncaric, F.; Feng, Y.; Gilbert, A.; Fernandes, J.F.; Bukhari, H.A.; Wajdan, A.; et al. The 'Digital Twin' to enable the vision of precision cardiology. European heart journal 2020, 41, 4556-4564, doi:10.1093/eurheartj/ehaa159.

[45].        Geris, L.; Lambrechts, T.; Carlier, A.; Papantoniou, I.J.C.O.i.B.E. The future is digital: in silico tissue engineering. 2018, 6, 92-98.

[46].        Yang, D.; Karimi, H.R.; Kaynak, O.; Yin, S. Developments of digital twin technologies in industrial, smart city and healthcare sectors: a survey. Complex Engineering Systems 2021, 1, 3, doi:10.20517/ces.2021.06.

[65].  Armeni, P.; Polat, I.; De Rossi, L.M.; Diaferia, L.; Meregalli, S.; Gatti, A. Digital Twins in Healthcare: Is It the Beginning of a New Era of Evidence-Based Medicine? A Critical Review. Journal of personalized medicine 2022, 12, doi:10.3390/jpm12081255.

[66].  Masison, J.; Beezley, J.; Mei, Y.; Ribeiro, H.; Knapp, A.C.; Sordo Vieira, L.; Adhikari, B.; Scindia, Y.; Grauer, M.; Helba, B.; et al. A modular computational framework for medical digital twins. Proceedings of the National Academy of Sciences of the United States of America 2021, 118, doi:10.1073/pnas.2024287118.

Round 2

Reviewer 5 Report (New Reviewer)

Thank you for the next opportunity to review the "The Digital Twin: A Potential Solution for Personalized Diagnosis and Treatment of Musculoskeletal System Diseases?”. In my opinion, the authors have corrected the manuscript in accordance with the recommendations. So I have no more comments and I think that the work should be accepted for publication.

Author Response

Thank you for your affirmation of our work.

This manuscript is a resubmission of an earlier submission. The following is a list of the peer review reports and author responses from that submission.

Round 1

Reviewer 1 Report

The paper can be interesting but need to be improved especially regarding the English level, which is very low.

The article is full of mistakes.

I strongly suggest a deep review of the English, after that the article could be considered.

Reviewer 2 Report

There is a lot written about digital twin (DT) healthcare and its promise. This perspective on the topic focuses on the musculoskeletal system. It is an interesting and important application. The authors introduce the topic with insights developed from their own work to implement the approach on this challenging system. They develop sufficient detail in the paper for the reader to appreciate both the advantages of the approach and complexity of the problems to implement the DT model. Practical limitations of DT as it applies to the musculoskeletal system that need to be addressed in the future are discussed. Overall, the authors present a realistic and useful picture for the technology that is promising and largely untapped. 

Reviewer 3 Report

The Digital Twin: an Potential Solution for Personalized Diagnosis and Treatment of Musculoskeletal Diseases is a perspective article aiming to suggest is suitability for the future. As a perspective study it is questionable how far the authors should go in their suggestions. In my opinion, there should be more caution in some statements.

Apart from that, it is clear that English needs to be deeply revised. There are too many flaws that distract the reader and make it difficult to understand.

Authors adopted “personality analysis”. Please revise as it is not an adequate term.

In the introduction, authors reinforce the role of biomechanics. For example, citing Cholewicki et al., 2019. Clearly the authors have not understand the counterpoint of this author, demonstrating that the present manuscript asks for clarification.

Finally, a perspective manuscript should be clear and concise. After reading and rereading this manuscript, I do consider it to confuse. Moreover, this type of article should be impartial. Every methodology have its advantages and disadvantages. It is clear that the authors are partial and only focus on the disadvantages of the conventional methods and the advantages of DT.